



# Estimation of depth-resolved profiles of soil thermal diffusivity from temperature time series and uncertainty quantification

Carlotta Brunetti[1], John Lamb[1], Stijn Wielandt[1], Sebastian Uhlemann[1], Ian Shirley[1], Patrick McClure[1], and Baptiste Dafflon[1]

[1]Lawrence Berkeley National Laboratory, Berkeley, CA 94720, USA

**Correspondence:** Carlotta Brunetti (brunetti.carlotta@gmail.com)

**Abstract.** Improving the quantification of soil thermal and physical properties is key to achieving a better understanding and prediction of soil hydro-biogeochemical processes and their responses to changes in atmospheric forcing. Obtaining such information at numerous locations and/or over time with conventional soil sampling is challenging. The increasing availability of low-cost, vertically resolved temperature sensor arrays offers promise for improving the estimation of soil thermal properties from temperature time series, and the possible indirect estimation of physical properties. Still, the reliability and limitations of such an approach needs to be assessed. In the present study, we develop a parameter estimation approach based on a combination of thermal modeling, sliding time-windows, Bayesian inference, and Markov chain Monte Carlo simulation to estimate thermal diffusivity and its uncertainty over time, at numerous locations and at an unprecedented vertical spatial resolution (i.e., down to 5 to 10 cm vertical resolution) from soil temperature time series. We provide the necessary framework to assess under which environmental conditions (soil temperature gradient, fluctuations, and trend), temperature sensor characteristics (bias and level of noise) and deployment geometries (sensor number and position) soil thermal diffusivity can be reliably inferred. We validate the method with synthetic experiments and field studies. The synthetic experiments show that in the presence of median diurnal fluctuations $\geq 1.5$ °C at 5 cm below the ground surface, temperature gradients $> 2$ °C m$^{-1}$, and a sliding time-window of at least 4 days, the proposed method provides reliable depth-resolved thermal diffusivity estimates with percentage errors $\leq 10\%$ and posterior relative standard deviations $\leq 5\%$ up to 1 m depth. Reliable thermal diffusivity under such environmental conditions also requires temperature sensors spaced precisely (with few-millimeter accuracy), with a level of noise $\leq 0.02$ °C, and with a bias defined by a standard deviation $\leq 0.01$ °C. Finally, the application of the developed approach to field data indicates significant repeatability in results and similarity with independent measurements, as well as promise in using a sliding time-window to estimate temporal changes in soil thermal diffusivity, as needed to potentially capture changes in carbon or water content.

## 1 Introduction

Knowledge of soil thermal properties (i.e., soil thermal conductivity, thermal diffusivity, and specific heat capacity) is fundamental to solving problems in many fields, such as in engineering, agriculture, meteorology, and geology (Farouki, 1981). Thermal properties, which are controlled by the spatial arrangement and fraction of soil components, such as minerals, or-



ganic matter, water, ice, and air, modulate heat fluxes in soil and at the soil-surface boundary. As a consequence, an improved quantification of soil thermal properties is a cornerstone for advancing the indirect estimation of fraction of soil components (Al Nakshabandi and Kohnke, 1965; Ochsner et al., 2001; Abu-Hamdeh, 2003; Arkhangel'skaya, 2009; Tong et al., 2016; Arkhangelskaya and Lukyashchenko, 2018; Xie et al., 2018; Jafarov et al., 2020) needed to parametrize hydro-biogeochemical models, as well as for understanding and predicting variability in subsurface thermal regime and fluxes of water, carbon, and

nutrients (Koven et al., 2013; Rasmussen et al., 2018; Nicolsky and Romanovsky, 2018; Oliva and Fritz, 2018).

Direct measurement of soil thermal and physical properties (Ochsner and Baker, 2008; Mengistu et al., 2017) obtained by sampling and analyzing soil from multiple depths and locations across the landscape is highly valuable, but it is also time-consuming, invasive, difficult to repeat over time, and costly. In addition, the sampling and analysis is not error free, primarily because of the difficulty in minimizing error in the bulk density measurement. Similarly, this approach provides limited vertical

and lateral resolution, which in some cases is insufficient to capture the highly heterogeneous distribution of soil physical and thermal properties. An assessment of alternative methods is needed that can potentially complement direct measurements and provide estimates of physical and thermal properties over space and time with unprecedented resolution. Soil temperature time series have been proven to be a valuable source of information for monitoring spatiotemporal changes in subsurface properties, and they have been used to infer (for instance) heat and water fluxes (e.g., Steele-Dunne et al., 2010; An et al., 2016; Irvine

et al., 2017; Tabbagh et al., 2017), organic matter content (Tran et al., 2016), and thermal properties (e.g., Nicolsky et al., 2009; Rajeev and Kodikara, 2016).

The estimation of thermal properties from soil temperature time series can be achieved through analytical or inverse methods. With analytical methods, thermal diffusivity is computed from the analytical solution of the heat equation, under the assumption that soil surface temperature is a sinusoidal function. In the literature, thermal diffusivity has been estimated from

the analytical solution of the heat conduction equation using, for instance, the Harmonic (Fourier) method (Carson, 1963; Beardsmore et al., 2020) and the popular phase-shift and amplitude-ratio methods (Krzeminska et al., 2012; Hinkel, 1997; Jong van Lier and Durigon, 2013; Andújar Márquez et al., 2016) or from the analytical solution of the conduction-convection equation (Gao et al., 2017, 2008). The main advantage of analytical methods is that they provide daily or monthly variation in bulk thermal diffusivity without the need for computational resources. These estimates are important input in physics-based

models that simulate land-atmosphere interactions (Gao et al., 2017). One limitation of these methods is that they cannot be applied when the temperature does not vary sinusoidally, such as in rainy or cloudy days. Moreover, since the amplitude of the soil temperature wave decreases exponentially with depth, the phase shift and amplitude methods might be unable to infer thermal diffusivity deeper than 20–30 cm (Farouki, 1981; Gao et al., 2017) unless very strong diurnal fluctuations are recorded at the surface of conductive soils. Additionally, most of the studies that rely on analytical methods are typically characterized

by a low vertical spatial resolution, which often involves the estimation of thermal properties in a single layer in the top 20 cm (e.g., Gao et al., 2017), in 2 layers in the top 30 cm (e.g., Jong van Lier and Durigon, 2013) or in 3 layers in the top 40 cm (Krzeminska et al., 2012; Gao et al., 2008). A few studies have investigated thermal properties at greater depths, such as up to 1 m with a resolution of 20 cm (Beardsmore et al., 2020) or with a single layer up to 5 m (Andújar Márquez et al., 2016).



Some of the challenges inherent to the analytical methods can be overcome by the use of numerical approximations of
the heat equation (that can take in input temperature time series of any form), and by embedding them in a deterministic
or probabilistic inverse method. A deterministic inversion relies on optimization techniques to estimate the value of the un-
known parameter(s) of interest (e.g., soil thermal properties) that minimize the difference between the simulated and measured
data (e.g., soil temperature time series). Deterministic inversion of thermal properties has been achieved with the finite dif-
ference (Krzeminska et al., 2012) or the finite element scheme to approximate the heat conduction (Kim et al., 2019), the
conduction-convection equation (Tabbagh et al., 2017), or the heat conduction equation with phase changes (Nicolsky et al.,
2009). Comparison of analytical and numerical methods can be found in Horton et al. (1983) and Rajeev and Kodikara (2016).
Deterministic inversions are computationally inexpensive, but the convergence to the global minimum is not guaranteed. Sim-
ilarly to the studies based on analytical methods, most of the investigations done with deterministic inverse approaches infer
thermal properties at a low vertical resolution, such as in a single layer (e.g., Tabbagh et al., 2017) or 2–3 layers in the top 40
cm (e.g., Kim et al., 2019; Krzeminska et al., 2012), or, for instance, 5 layers up to 50 m (Nicolsky et al., 2009). Moreover,
deterministic inversion, as with analytical methods, only provides a single solution of the system, with no means to explicitly
model and take into account different sources of uncertainty.

Probabilistic inverse methods provide a set (distribution) of equally probable values of the unknown parameters by aiming
at "capturing both the average response of the system and the variability due to uncertainties of any kind" (Renard et al., 2013).
In the last decades, Bayesian inference has gained popularity among the probabilistic inverse approaches. Unlike classical
statistical methods to estimate uncertainty (e.g., Beardsmore et al., 2020), Bayesian inference allows incorporating the prior
knowledge about the parameter to be estimated as well as taking explicitly into account different sources of uncertainty,
such as measurements errors. Moreover, a great variety of sampling algorithms are available to avoid nonconvergence issues.
Though promising, the applications of the Bayesian framework to infer soil thermal properties are still limited. Huang et al.
(2017) have derived the heat conductivity in the layer between the atmosphere and the soil surface, Choi et al. (2018) have
inferred soil thermal conductivity from a thermal response test, and Tran et al. (2016) have inferred organic matter content from
soil temperature, liquid water, and apparent resistivity data. Bayesian inference has been more widely applied in engineering
(Kaipio and Fox, 2011) to estimate thermal properties of fins (Gnanasekaran and Balaji, 2013; Somasundharam and Reddy,
2017) and walls (De Simon et al., 2018; Rodler et al., 2019). While the above studies have shown promise in estimating
thermal properties from time series of temperature, several challenges still remain, including the need for an approach that can
assess the conditions under which such estimates are reliable. In addition, the recent increasing availability of dense, vertically
resolved arrays of temperature sensors offer new opportunity to increase the vertical and lateral resolution of the estimates and
potentially quantify their temporal variability, although such a strategy has not been assessed yet.

The aim of our study is to evaluate the potential of estimating soil thermal diffusivity and its uncertainty at numerous loca-
tions across the landscape and at an unprecedented vertical spatial resolution (i.e., 5–10 cm in the top 1 m), using vertically
resolved time series of soil temperature. A second objective is to explore the possibility of estimating changes in thermal
diffusivity over time by sequencing the soil temperature time series with a sliding time-window. To this end, we developed
a parameter-estimation approach based on Bayesian inference that also allows us to assess the impact of different sources of



uncertainty linked to various environmental conditions (e.g., soil temperature gradient, fluctuations, and trend), temperature
sensor characteristics, and deployment geometries. Hence, this study aims at answering the following research questions: (1)
Can soil thermal diffusivity be reliably estimated (i.e., with percentage errors $\leq 10\%$ and posterior relative standard deviations
$\leq 5\%$) at multiple depths and locations from solely vertically resolved soil temperature time series (i.e., without additional
data), and if so, under which environmental conditions and sensor characteristics? (2) Can we capture temporal changes in
thermal diffusivity by estimating diffusivity sequentially for a sliding time-window of soil temperature time series? (3) Can
thermal diffusivity, estimated by applying our method to several locations across the landscape, be used to retrieve soil com-
position and other soil thermal-physical properties?

To address these questions, we perform synthetic experiments in which we infer soil thermal diffusivity and assess its
uncertainty under different soil temperature gradients and fluctuations, length of sliding time-window, level of measurement
errors, and temperature sensor geometries (Sect. 3). In Sect. 4, we further evaluate the reliability and sensitivity of the proposed
method with an in situ study that compares estimated thermal diffusivities for a silty/clayey soil (Berkeley, California, USA)
with independent measurements obtained with a thermal-properties analyzer. Finally, in Sect. 5, we apply the method at a field
site in a discontinuous permafrost environment (Nome, Alaska, USA), and compare thermal diffusivity estimates at numerous
locations across the site with soil sample measurements.

## 2 Theory and method

### 2.1 Heat equation and thermal diffusivity

Heat is exchanged at the soil surface and within the soil through different processes, such as radiation, convection, conduction,
and latent heat. However, conduction is the process that dominates the transport of heat in soil (Hillel, 1982). Assuming a purely
heat-conduction process in a heterogeneous medium of length $L$ [m] under unsteady-state conditions over a time-window of
duration $T$ [s], the heat equation in one dimension is (Hillel, 1982):

$$\frac{\partial u}{\partial t} = \frac{\partial}{\partial z}\left(\alpha \frac{\partial u}{\partial z}\right) \qquad z \in (0.05, L), t \in (0, T] \tag{1}$$

where temperature, $u$ [°C], varies with time, $t$ [s], and depth, $z$ [m], and thermal diffusivity, $\alpha$ [m s$^{-2}$], changes with $z$. Thermal
diffusivity characterizes unsteady heat conduction and defines how quickly a material transfers heat from a hot to a cold area
due to a change in temperature (Farouki, 1981). High thermal diffusivity values indicate that a material is capable of a rapid
transfer of heat. Thermal diffusivity is defined as the ratio of the thermal conductivity, $\kappa$ [W m$^{-1}$ K$^{-1}$], to the density, $\rho$ [kg
m$^{-3}$], and specific heat capacity, $c_p$ [J kg$^{-1}$ K$^{-1}$] at a constant pressure:

$$\alpha = \frac{\kappa}{\rho \cdot c_p} \tag{2}$$

Thermal diffusivity of soil is influenced by (1) soil composition, (2) soil water content, (3) soil bulk density (that depends
mainly on the soil composition and its degree of compaction), and, to a lesser extent (4) temperature changes (Al Nakshabandi



and Kohnke, 1965; Ochsner et al., 2001; Abu-Hamdeh, 2003; Arkhangel'skaya, 2009; Tong et al., 2016; Arkhangelskaya and
Lukyashchenko, 2018; Xie et al., 2018; Zhu et al., 2019).

## 2.2   Forward modeling: finite differences and sliding time-windows

The differential equation in Eq. (1) can be solved numerically by discretizing the space-time domain via finite difference or
finite element methods such that an approximation of the exact solution at the grid nodes can be computed. We apply herein the
most common and easy-to-implement explicit finite difference method based on the forward time and centered space scheme
(Petter Langtangen and Linge, 2017; Praprotnik et al., 2004):

$$u_i^{j+1} = u_i^j + \frac{\Delta t}{\Delta z^2} \left[ \alpha_{i+\frac{1}{2}} (u_{i+1}^j - u_i^j) - \alpha_{i-\frac{1}{2}} (u_i^j - u_{i-1}^j) \right] \tag{3}$$

where $i$ and $j$ identify the grid node in space and time, respectively; $\Delta t$ [s] is the length of the time step; $\Delta z$ [m] is the
distance in space between grid nodes; and $u_i^j$ represents a discrete approximation to $u(z,t)$. Note that the thermal diffusivity is
computed on a staggered grid. Both $\Delta t$ and $\Delta z$ are chosen so as to avoid numerical instabilities and to ensure the convergence
of the explicit finite difference scheme:

$$\Delta t \leq \frac{\Delta z^2}{2 \cdot \max(\alpha)} \tag{4}$$

which is known as the Courant-Friedrichs-Levyor condition (Courant et al., 1928). We consider thermal diffusivity values as
large as $\alpha = 3 \cdot 10^{-6}$ m$^2$ s$^{-1}$ and $\Delta z = 0.05$ m; therefore, we set $\Delta t = 300$ s. While a uniform spacing of 0.05 m is used in the
finite difference scheme, the actual spatial resolution at which soil thermal diffusivity is resolved by our method corresponds
to the spacing between the temperature sensors. Similarly to Tabbagh et al. (2017) and Rodler et al. (2019), we sequence the
soil temperature time series with a sliding time-window of length $T$ that moves every 24 hours over the entire time period
for which data are available. Thermal diffusivity is assumed to be constant in time within each time-window. The optimal
time-window length $T$ is investigated in detail later in this study, as it needs to be sufficiently small to limit the influence
of hydrological processes (e.g., advection not represented in the heat-conduction-based model), but long enough to contain
enough information to reliably infer thermal diffusivity at multiple depths. Based on the measurement geometry assumed in
this study, the soil temperature values at multiple depths at the beginning of each time-window are used as initial conditions,
whereas the soil temperature time series recorded over the time-window by the shallowest and deepest sensors are employed
as top and bottom boundary conditions (i.e., time-varying Dirichlet boundary conditions), respectively.

## 2.3   Inverse modeling: Bayesian inference with MCMC

Bayesian inference provides a probabilistic framework that enables us to derive from $n$ data, $\widetilde{Y}=\{\widetilde{y}_1,...,\widetilde{y}_n\}$, the $d$-dimensional
vector of the parameters of interest, $\boldsymbol{\theta}$, that are not directly measured or known while fully quantifying the associated uncer-
tainty. This process is performed by the Bayes'theorem:

$$p(\boldsymbol{\theta}|\widetilde{\boldsymbol{Y}}) = \frac{p(\boldsymbol{\theta})p(\widetilde{\boldsymbol{Y}}|\boldsymbol{\theta})}{p(\widetilde{\boldsymbol{Y}})} \tag{5}$$





that defines how the prior state of knowledge about the quantities of interest, $p(\boldsymbol{\theta})$, is updated by the information contained in the data through the likelihood function, $p(\widetilde{\boldsymbol{Y}}|\boldsymbol{\theta})$. The denominator in Eq. (5) is a normalization factor called "evidence" that can be neglected if a single conceptual model is considered, as in this study. The Bayes' theorem provides as output the posterior probability density function (pdf), $p(\boldsymbol{\theta}|\widetilde{\boldsymbol{Y}})$, of the parameters of interest $\boldsymbol{\theta}$. Bayesian inference can be performed with likelihood functions of any form. However, we assume here, as it is often the case, uncorrelated and normally distributed measurement errors with constant standard deviation, $\sigma_{\widetilde{\boldsymbol{Y}}}$, that define a Gaussian likelihood function as:

$$p(\widetilde{\boldsymbol{Y}}|\boldsymbol{\theta}) = \left(\sqrt{2\pi\sigma_{\widetilde{\boldsymbol{Y}}}^2}\right)^{-n} \exp\left[-\frac{1}{2}\sum_{h=1}^{n}\left(\frac{\mathcal{F}_h(\boldsymbol{\theta}) - \widetilde{y}_h}{\sigma_{\widetilde{\boldsymbol{Y}}}}\right)^2\right]. \tag{6}$$

The term $\mathcal{F}_h(\boldsymbol{\theta})$ is the forward model (e.g., Eq. (3), Sect. 2.2) used to simulate the observed data, $\widetilde{\boldsymbol{Y}}$. Larger likelihood values indicate that $\mathcal{F}_h(\boldsymbol{\theta})$ better predicts the data at hand. The posterior pdf in Eq. (5) is multidimensional and analytically intractable. Sampling schemes such as the popular Markov chain Monte Carlo (MCMC) (Gilks et al., 1995; Robert and Casella, 2013) algorithm are, therefore, applied to find an approximation of the posterior pdf. In particular, in this work, we make use of the DiffeRential Evolution Adaptive Metropolis, DREAM$_{(ZS)}$, algorithm (Laloy and Vrugt, 2012; Vrugt, 2016) that is a multi-chain MCMC sampling scheme based on the Metropolis acceptance ratio (Metropolis et al., 1953) but with an improved sampling efficiency. This algorithm allows us to avoid nonconvergence issues that might arise when inferring for more than one unknown parameter (i.e., thermal diffusivity at multiple depths). The convergence of the multiple Markov chains to the posterior pdf is assessed quantitatively with the Gelman-Rubin statistic (Gelman et al., 1992). In this paper, we aim at inferring about 12 unknown parameters; therefore, following the guidelines provided by Vrugt (2016), we set the number of Markov chains to 3 and a total of $5 \cdot 10^4$ iterations were found to be enough to reach convergence and get acceptable acceptance rates (i.e., between 15% and 40%, according to Gelman et al. (1996)). The unknown parameters to be inferred from the soil temperature time series are thermal diffusivity values at multiple depths that are drawn from the uniform prior distribution U[0.01, 3] mm$^2$s$^{-1}$. The upper and lower limits of the thermal diffusivity prior range have been chosen based on typical values found in the literature (e.g., Farouki (1981); Andújar Márquez et al. (2016)). In the case of the in situ experiment (Sect. 4) and field case study (Sect. 5), we also infer the standard deviation of the measurement errors, $\sigma_{\widetilde{\boldsymbol{Y}}}$, by drawing it from the uniform distribution U[0, 1] °C.

The uncertainty over the MCMC thermal diffusivity estimates is quantified with the percentage error and the posterior relative standard deviation. The percentage error (PE) is the distance of the MCMC posterior mean of thermal diffusivity, $\overline{\alpha}$, from the synthetic true value, $\alpha_{true}$, (in the case of synthetic experiments):

$$PE = \frac{|\overline{\alpha} - \alpha_{true}|}{\alpha_{true}} \cdot 100\% \tag{7}$$

whereas the posterior relative standard deviation (RSD) measures the dispersion of the MCMC posterior distribution around the MCMC posterior mean of thermal diffusivity:

$$RSD = \frac{\sigma_\alpha}{\overline{\alpha}_{mcmc}} \cdot 100\% \tag{8}$$

where $\sigma_\alpha$ is the standard deviation of the posterior MCMC distribution of thermal diffusivity.



The uncertainty regarding the inferred thermal diffusivity at multiple depths is influenced by a combination of many factors, including the amount of temperature gradient at each depth, the length of the sliding time-window used in the MCMC inversion, the amount of information contained in temperature trend and fluctuations, the level of noise and bias in the temperature time series, and the geometry of temperature sensors.

## 2.4    Data and field sites description

### 2.4.1    Vertically resolved time series of temperature

Synthetic and field experiments performed in this study use an acquisition geometry in which temperature measurements are collected autonomously at numerous depths, with a vertical resolution of 0.05 m to 0.1 m. This geometry corresponds to measurements obtained with vertically resolved temperature devices such as, for example, a Distributed Temperature Profiling (DTP) system (Dafflon et al., 2021). In this study, each DTP system has at least 13 sensors that provide temperature time series

at a spatial resolution of 0.05 m between the top 5 sensors, and of 0.10 m between the bottom 8 sensors. The temperature probe is typically inserted in the subsurface such that the sensors are at 0.05, 0.10, 0.15, 0.20 0.25, 0.35, 0.45, 0.55, 0.65, 0.75, 0.85, 0.95 and 1.05 m below the ground surface. The sensors record the soil temperature every 15 min, with a resolution of 0.0078 °C and an accuracy (corresponding to three standard deviations) of ± 0.1 °C or ± 0.015 °C when relying on the manufacturer calibration or an additional in house calibration, respectively.

### 2.4.2    Synthetic experiment

The synthetic experiments (Sect. 3) are implemented to investigate the impact of different environmental conditions and sensor characteristics on the uncertainty of thermal diffusivity estimates. We generate synthetic temperature fields that represent various types of temperature gradients and fluctuations. This is achieved through forward modeling (Sect. 2.2) with initial, top, and bottom boundary conditions set equal to the temperature time series observed at a monitoring site in Alaska during summer

(Romanovsky et al., 2020) and autumn (data from this study), and by assuming a soil column composed of three layers (i.e., top layer at 0.05–0.1 m, middle layer at 0.1–0.42 m, and bottom layer at 0.42–1.05 m). The thermal diffusivity in the three layers is assumed to be constant over time and equal to 0.16, 0.27 and 0.43 mm$^2$s$^{-1}$ for the case of summer temperatures and 0.25, 0.75 and 0.6 mm$^2$s$^{-1}$ for autumn. The summer case is characterized by median diurnal fluctuations (maximum minus minimum temperature over one day measured by the top sensor at 0.05 m below the ground surface) of 1.6 °C over the time

period considered and temperature gradients all larger than 2 °C m$^{-1}$ at each depth. The case for autumn has lower median diurnal fluctuations (i.e., 0.13 °C) and lower temperature gradients (mostly within -1 °C m$^{-1}$ and 2 °C m$^{-1}$). We perturb the synthetic temperature fields using a Gaussian noise with a standard deviation of $\sigma_{\widetilde{Y}}$ = 0.02 °C.

### 2.4.3    Assessment of in situ estimations of thermal diffusivity

One of the two field experiments presented in this study is aimed at evaluating the repeatability of the estimated diffusivity

and the method's ability to detect changes over time (Sect. 4). To this end, measurements from four co-located (in a 25 cm





radius area) DTP systems installed from 23 December 2020 to 17 February 2021, in a silt-dominated soil in Berkeley (CA, USA), were used to estimate thermal diffusivity. In this winter period, the median [min,max] temperature gradient among all the probes recorded between the two top sensors is 5.98 °C m$^{-1}$ [-25.59 °C m$^{-1}$, 37.07 °C m$^{-1}$]; between the two bottom sensors it is 2.16 °C m$^{-1}$ [1.16 °C m$^{-1}$, 2.91 °C m$^{-1}$]. The median [min,max] diurnal temperature variation measured by the top sensor at 0.05 m below the ground surface is 1.59 °C [0.52 °C, 3.67 °C]. Accumulated precipitation every 15 min recorded from the Lawrence Berkeley National Laboratory meteorological station (lat:37.8771, long:-122.2486) was obtained from https://mesowest.utah.edu. Independent measurements of thermal diffusivity were collected on 24 December 2020, after a long dry period that lasted about 30 days, using a thermal properties analyzer (TEMPOS instrument with the SH-3 dual needle; METER Group), which measures conductivity, diffusivity, and volumetric heat capacity with an accuracy of ±10%. Five measurements were recorded at each 0.10, 0.20, 0.30, 0.40, 0.50 and 0.55 m depth, by repetitively inserting the SH-3 dual needle along the walls of a hand-augered 8 cm diameter hole.

### 2.4.4 Field study in a discontinuous permafrost environment

The second field experiment presented in this study involves the measurement and estimation of soil thermal diffusivity at numerous locations in a discontinuous permafrost environment, and the evaluation of the links between the estimated soil thermal diffusivity values and soil physical properties (Sect. 5). The study site is located along Teller Road about 40 km northwest of Nome, Alaska (64.72 ° N, 165.94 ° W). This site, referred to here as the "Teller site", is characterized by discontinuous permafrost and a great variety of vegetation types (e.g., tall shrub, dwarf shrub, moss, graminoids (Léger et al., 2019)). Soil thermal diffusivity was inferred from temperature time series recorded in the dry period 7–27 October 2019 from 27 locations where soil was entirely unfrozen (i.e., temperature time series at all depths above 0.5 °C). At this time of the year, the median [min,max] temperature gradient among all the probes recorded between the two top sensors was 10.47 °C m$^{-1}$ [-9.22 °C m$^{-1}$, 35.78 °C m$^{-1}$], and between the two bottom sensors was as low as 0.63 °C m$^{-1}$ [-0.94 °C m$^{-1}$, 2.19 °C m$^{-1}$]. The median [min,max] diurnal temperature variation measured by the top sensor at 0.05 m below the ground surface was 0.23 °C [0.03 °C, 1.41 °C].

Moreover, 92 soil samples at 50 locations were retrieved during the first week of August 2019 and analyzed in the laboratory in order to measure thermal and physical properties. Thermal conductivity, thermal diffusivity, volumetric heat capacity, and wet bulk density were measured a few hours after the soil samples were collected, using a weighing scale and the thermal properties analyzer. Later analyses in the lab provided dry bulk density, water content, and carbon density. A total of 20 of these soil samples, retrieved at about 0.075 m, 0.20 m and 0.80 m depth, were collocated, with 13 out of the 27 locations monitored with the DTP systems.

### 3 Method assessment through synthetic experiments

Synthetic experiments were performed to investigate how the uncertainty of the inferred thermal diffusivity at multiple depths is affected by various factors, including (1) the amount of temperature gradient at each depth (Sect. 3.1 and 3.3), (2) the length





of the sliding time-window used in the MCMC inversion (Sect. 3.1), (3) the amount of information contained in temperature trends and fluctuations (Sect. 3.2), (4) the level of noise and bias in the temperature time series (Sect. 3.3), and (5) the geometry

of the temperature sensors (Sect. 3.4).

## 3.1    Impact of time-window length

In this section, we assess how the uncertainty of thermal diffusivity estimates is affected by the length of the sliding time-window used in the MCMC inversion under two very different types of temperature fields. One temperature field has a strong gradient and diurnal fluctuations that generally occur in summer (Fig. 1a), whereas the other has much more limited gradient

and fluctuations that are more typical for the autumn season (Fig. 1h). The diurnal signal in the top boundary condition in summer allows us to get PE ≤ 10% and posterior RSD at maximum ±5% on thermal diffusivity estimates in the top 0.25 m, even for very short time-windows of 1 day (Fig. 1b, 1f, 1g). Using a time-window of four or more days allows us to infer thermal diffusivity with a PE ≤ 10% and a RSD of ±2% at all depths up to 1 m (Fig. 1c, 1f, 1g). A lower diurnal signal in the top boundary condition (between -1 °C m$^{-1}$ and 2 °C m$^{-1}$), as observed in autumn (Fig. 1h), implies much larger error.

Indeed, in this case a time-window of at least 21 days is required in order to infer thermal diffusivity at all depths with a PE ≤ 10% and a RSD of ±5% (Fig. 1k, 1m, 1n). In general, thermal diffusivity estimates in the bottom part of the soil column, where temperature gradients and diurnal fluctuations are smaller than in the upper part, are characterized by higher uncertainty.

This synthetic experiment further shows that soil heterogeneity is defined at a spatial resolution equal to the spacing of the temperature sensors. When an interface is located between two sensors, such as the one at 0.42 m in our synthetic soil column,

the bulk thermal diffusivity inferred between 0.35 m and 0.45 m corresponds to the weighted mean of the thermal diffusivity above and below the soil interface.

## 3.2    Impact of temperature trend and fluctuations

In this section, we assess the effect of temperature trend versus diurnal fluctuations over time (Fig. 2a–d) on the uncertainty of the estimated thermal diffusivity. For this purpose, we perform the MCMC inversion on various cases, which include (F0)

a summer synthetic temperature field (Fig. 2a), (F1) a detrended summer synthetic temperature field (Fig. 2b), (F2) a summer synthetic temperature field with daily and smaller fluctuations smoothed out (Fig. 2c), and (F3) a summer synthetic temperature field without any fluctuations (Fig. 2d). All four cases preserve the same vertical temperature gradient, with a sliding time-window of 7 days.

Results indicate that the information contained in the temperature trend has negligible value for estimating soil thermal

diffusivity compared to the information provided by the fluctuations (F0 vs F1, Fig. 2e–f). Indeed, the uncertainty of the thermal diffusivity estimates does not worsen when detrending the time series at each depth. Smoothing out the diurnal and smaller fluctuations can result in percentage errors higher than 10% (F2 in Fig. 2e) and higher than 30% when completely removing them (F3 in Fig. 2e). However, we should highlight that we can still get reliable thermal diffusivity estimates (PE ≤ 10%, RSD ≤ 1%) in F2 and F3 by increasing the time-window length from 7 to 20 days (F2b and F3b in Fig. 2e–f).



Earth **Surface**
**Dynamics**
Discussions

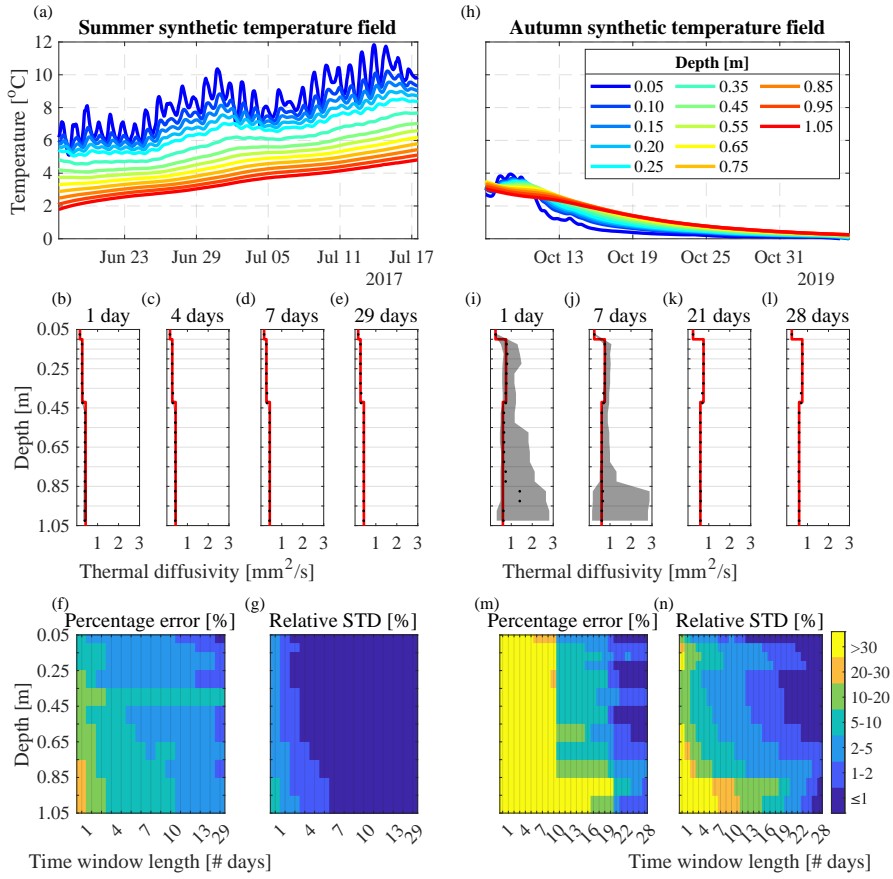

**Figure 1.** Synthetic temperature field for (a) summer and (h) autumn and corresponding MCMC inversion results for (b–g) summer and (i–n) autumn. (b–e, i–l) Median (black dots) and range (black shaded areas) of the posterior means of thermal diffusivity at each depth derived from MCMC simulation for various sliding time-window lengths. Red lines show the "true" thermal diffusivity profile. (f–g, m–n) Percentage error (Eq. (7)) and posterior relative standard deviation (Eq. (8)) at each depth for sliding time-windows of different lengths (x-axis). Note that the time-window length in the x-axis is nonlinear.

## 3.3 Impact of measurement noise and bias

In this section, we investigate the impact of temperature sensor noise and bias on inferred thermal diffusivity estimates. We consider two distinct scenarios in which temperature data are perturbed with uncorrelated and normally distributed measurement errors, by using different combinations of the mean, $\mu$, and standard deviation, $\sigma_{\widetilde{Y}}$. In the first scenario, soil temperature time series are assumed to be measured by temperature probes with unbiased sensors, $\mu = 0$ °C, that have different levels of noise $\sigma_{\widetilde{Y}}$ equal to 0.01 °C, 0.02 °C, 0.03 °C, 0.05 °C, and 0.1 °C. The second scenario considers low levels of noise in the temperature time series, $\sigma_{\widetilde{Y}} = 0.01$ °C, and evaluates sensor bias, so that $\mu$ is randomly drawn from a zero-mean Gaussian



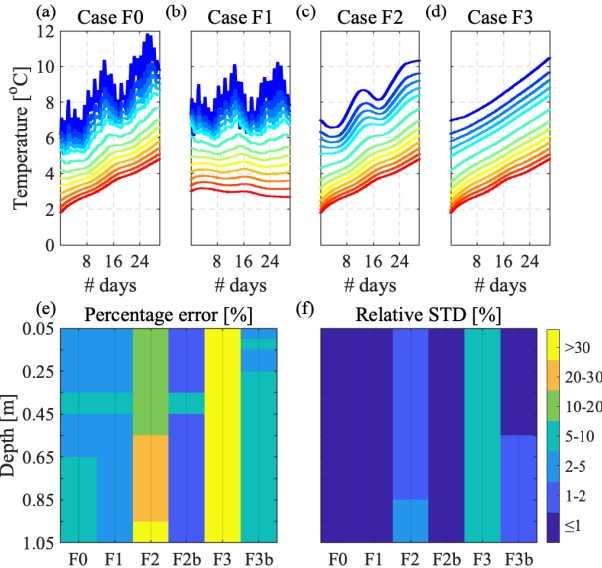

**Figure 2.** Impact of soil temperature trend and fluctuations on the MCMC inversion evaluated for various cases, including (a) summer trend and fluctuations (F0), (b) detrended fluctuations (F1), (c) smoothed out daily and smaller fluctuations (F2), and (d) without fluctuations (F3). (e) Percentage error (Eq. (7)) and (f) posterior relative standard deviation (Eq. (8)) at each depth when using sliding time-windows of 7 days for all four temperature fields (F0–F3) and of 20 days (F2b and F3b) for temperature field F2 and F3.

distribution with standard deviation, $\sigma_c$, set equal to 0.005 °C, 0.01 °C, 0.02 °C and 0.04 °C. The MCMC inversion results (Fig. 3a–l) when using the summer temperature data of Fig. 1a show that thermal diffusivity can be inferred with PEs $\leq$ 10% (Fig. 3e) and posterior RSDs $\leq$ 1% (Fig. 3f) when the noise $\sigma_{\widetilde{Y}}$ is no larger than 0.02 °C. The percent error may increase up to 20% with $\sigma_{\widetilde{Y}}$=0.05 °C. Bias in the temperature sensors has an overall larger impact on the PE of thermal diffusivity than a high level of noise in temperature sensors (Fig. 3g–j vs 3a–d). Indeed, even with biases applied at each depth drawn randomly from a Gaussian distribution with $\sigma_c$ as low as 0.01 °C, we can observe biased thermal diffusivity estimates (Fig. 3h) with PE > 10% (Fig. 3k), and this effect worsens when increasing the bias in the temperature sensors (Fig. 3j).

## 3.4 Impact of sensor geometry

In this section, we evaluate how thermal diffusivity estimates are affected by the temperature sensor geometry, including the number and position of the sensors, and the effect of potential mispositioning of the sensors in soil. The sensor spatial configuration of the temperature probe used in the present work and described in Sect. 2.4.1 is considered the reference case and is indicated with P0 in Fig. 4.

When increasing the sensor spacing from 0.05 m to 0.1 m in the top 0.25 m (P1), the PE of thermal diffusivity estimates can be as high as 20% in the top 0.1 m, which is likely strengthened by the presence of an interface between the two top sensors





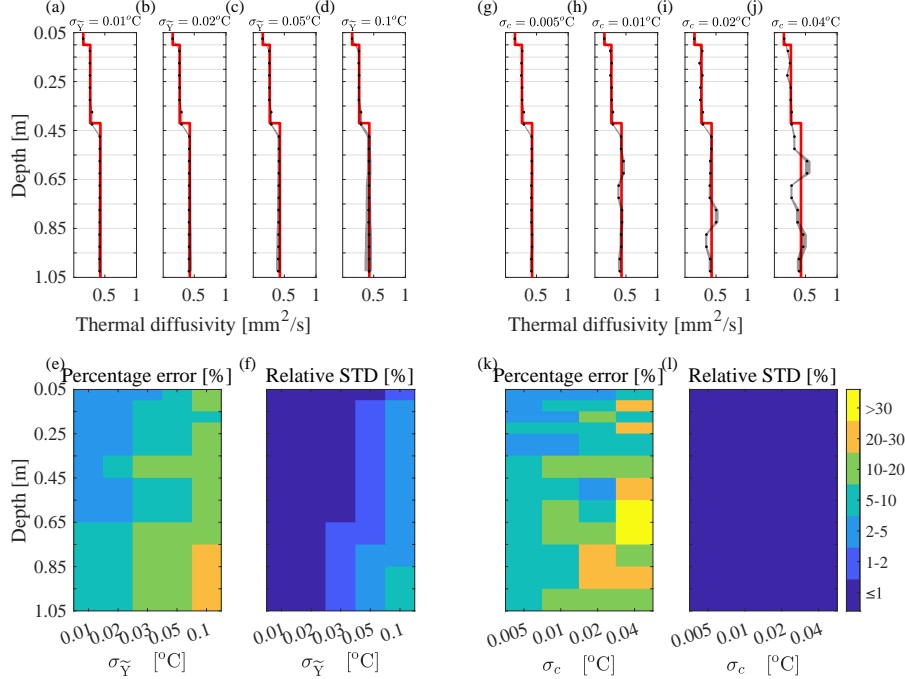

**Figure 3.** MCMC inversion results for the synthetic temperature field in Fig. 1a when perturbed by (a–f) Gaussian noise with different standard deviations, $\sigma_{\widetilde{Y}}$, and (g–l) bias applied at each depth, drawn randomly from a Gaussian distribution with different standard deviations, $\sigma_c$. (a–d, g–j) Median (black dots) and range (black shaded areas) of the posterior means of thermal diffusivity at each depth is derived from MCMC simulation in each sliding time-window of 7 days. Red lines show the "true" thermal diffusivity profile. Percentage error (Eq. (7)) and posterior relative standard deviation (Eq. (8)) at each depth when applying (e–f) different level of noise and (k–l) different biases in the temperature data used in the MCMC inversion.

(P0 vs P1 in Fig. 4a). Moreover, a lower resolution in the top part of the probe increases the PE of the estimates in the bottom part (P1 in Fig. 4a). Increasing the sensor spacing from 0.1 m to 0.2 m in the bottom part of the probe (i.e., below 0.4 m deep), where the soil is homogenous, does not significantly affect the PE of the thermal diffusivity estimates (P2 in Fig. 4a). A sensor spacing of 0.2 m along the entire probe yields PE larger than 10% and $\leq 20\%$ (P3 in Fig. 4a). The sensor geometries in P0,

P1, P2, and P3 do not significantly impact the RSDs of the estimates, which are $\leq 2\%$ (Fig. 4b). Decreasing the number of sensors to 3 or 4 leads to unreliable thermal diffusivity estimates (P4 and P5 in Fig. 4a), and the corresponding posterior RSDs are higher (Fig. 4b). Indeed, geometries with a significantly smaller number of sensors, such as probe P4 and P5, rely on much less data to compute the likelihood and constrain the MCMC inversion.

We evaluate further how an error in positioning a sensor at a specific depth or at a specific distance from other sensors

impacts the estimated thermal diffusivity. Note that this risk is more strongly present when using discrete temperature sensors placed manually in the subsurface (Fig. 4c–d) instead of a distributed temperature profiling system. A downwards shift of 0.02 m at the top sensor leads to unreliable thermal diffusivity estimates in the uppermost part of the soil column, and to PEs on





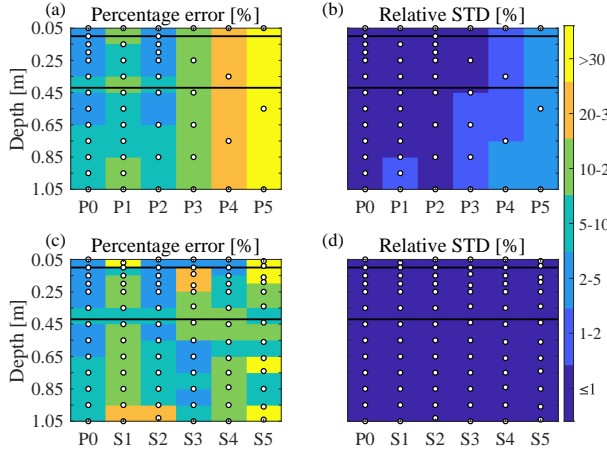

**Figure 4.** MCMC inversion results with a 7 days sliding time-window to evaluate the impact of different temperature sensor geometries. (a) Percentage error (Eq. (7)) and (b) relative standard deviation (Eq. (8)) at each depth when using different number and position of the temperature sensors (white dots). (c) Percentage error (Eq. (7)) and (d) relative standard deviation (Eq. (8)) at each depth when considering different potential mispositioning of the temperature sensors (white dots). The black bold horizontal lines depict the position of soil interfaces in the "true" synthetic thermal diffusivity profile.

thermal diffusivity larger than 10% over the entire soil column (case S1 in Fig. 4c). Shifting the bottom sensor 0.02 m upwards has a smaller impact on the PE of thermal diffusivity estimates (case S2 in Fig. 4c). The main reasons for this are that the soil
is homogeneous in the bottom part of the soil column below 0.42 m, and that thermal diffusivity is more sensitive to changes in the top temperature time series compared to the bottom ones. Indeed, the top sensors record the temperature time series with the highest content of information (e.g., diurnal and seasonal fluctuations). Similarly, applying a shift at each sensor above 0.5 m depth, randomly drawn from the set -0.01, 0, 0.01 m, degrades the PE of thermal diffusivity estimates more strongly than when the shifts are applied to the sensors below 0.5 m depth (case S3 vs S4 in Fig. 4c). If the shifts are applied to all sensors, the
PEs are higher than 30% (case S5 in Fig. 4c). Reducing the number of sensors leads to higher uncertainty in thermal diffusivity estimates (i.e., increase in posterior RSDs), because less data is used to constrain the inversion. Instead, a malpositioning of one or more sensors does not affect the RSD of thermal diffusivity estimates (i.e., all $\leq$ 1%, Fig. 4d), since the number of sensors and, hence, data used in the inversion does not change. Moreover, we find that increasing the sensor spacing from 5 cm to 10 cm in the top 25 cm of soil (probe P1 in Fig. 4a) has a smaller impact on the PE of thermal diffusivity estimates than an
error of a few centimeters in positioning these sensors (case S1 and S3 in Fig. 4c).

## 4   Assessment of in situ estimations of thermal diffusivity

The estimation of soil thermal diffusivity from temperature time series was assessed by comparing MCMC inversion results from multiple depth-profiles of temperature located close to each other, with independent measurements obtained using the





thermal properties analyzer (Sect. 2.4.3). The potential of the MCMC method to infer thermal diffusivity over time was also
further evaluated using the sliding time-window approach over a two-month period.

First, we compared the thermal diffusivity estimated with the developed method to those measured with the thermal prop-
erties analyzer. The replicated measurements from the thermal properties analyzer, while showing a high variability (large
black error bars in Fig. 5a), suggest a fairly homogenous clayey/silty soil, which is in agreement with visual observations and
values found in the literature (e.g., Andújar Márquez et al., 2016). The thermal diffusivity inferred from the temperature time
series show very similar values, although one of the probes (Probe 4) provides slightly higher values than the others at depth
greater than 30 cm. Both the measurements from the thermal properties analyzer and the MCMC-inferred thermal diffusivity
estimates show similar values and trend in the top 30 cm. Deeper, the measurements from the thermal analyzer suggest a slight
decrease in thermal diffusivity that is not visible in the MCMC estimates. The thermal diffusivity values are still comparable,
considering that their variability is within the range of typical clay/silt soils, and that both methods tend to show the presence
of spatial variability in soil thermal properties.

Further, we evaluated the variation in thermal diffusivity over time estimated using the sliding time-window approach. The
MCMC inversion results from temperature data recorded by the four temperature probes are very similar and show that the
estimated thermal diffusivity values in the top 30 cm of soil remain consistently lower than at deeper depths over the entire
2-months period. While the top soil does not show much temporal variability in the soil diffusivity over time, the deeper part
shows larger changes, particularly at the time of precipitation events occurring between 22 January and 5 February 2021. These
precipitations events are associated with a decrease in temperature (Fig. 5b) and an increase in thermal diffusivity, mainly in
the bottom part of the soil column (Fig. 5c). Thermal diffusivities inferred from the sliding time-window associated with the
large precipitation events show posterior RSDs that are more than double those obtained from the MCMC inversion in the
other time-windows (Fig. 5d). Moreover, the MCMC inversion provides a worse fit to the temperature data in the time-window
corresponding to large precipitation events (Fig. 5e). Indeed, we find that the inferred standard deviation of the temperature
measurement errors, $\sigma_{\widetilde{Y}}$, (Sect. 2.4.3) is typically within 0.01 °C and 0.02 °C, but increases to 0.06 °C between 22 January and
5 February 2021 (Fig. 5e). Despite this increase in uncertainty in the bottom part of the soil column during the precipitation
events, the observed changes in thermal diffusivity at these depths is consistent with the expected impact of an increase in water
content in an initially relatively dry soil (e.g., Farouki, 1981; Arkhangelskaya and Lukyashchenko, 2018) . Unfortunately, we
do not have soil moisture data and water level measurements to further investigate the controls on these changes in thermal
diffusivity.

## 5   Estimation of soil thermal diffusivity in a discontinuous permafrost system

The MCMC method is used to estimate unfrozen soil thermal diffusivity at multiple depths and locations across a discontinuous
permafrost environment along Teller Road (AK). Improving the estimation of thermal diffusivity in such an environment is
critical to potentially improving the estimation of soil physical properties, including organic matter concentration and bulk
density, that critically influence carbon cycle dynamics. A particular goal of this study is to infer soil thermal diffusivity at



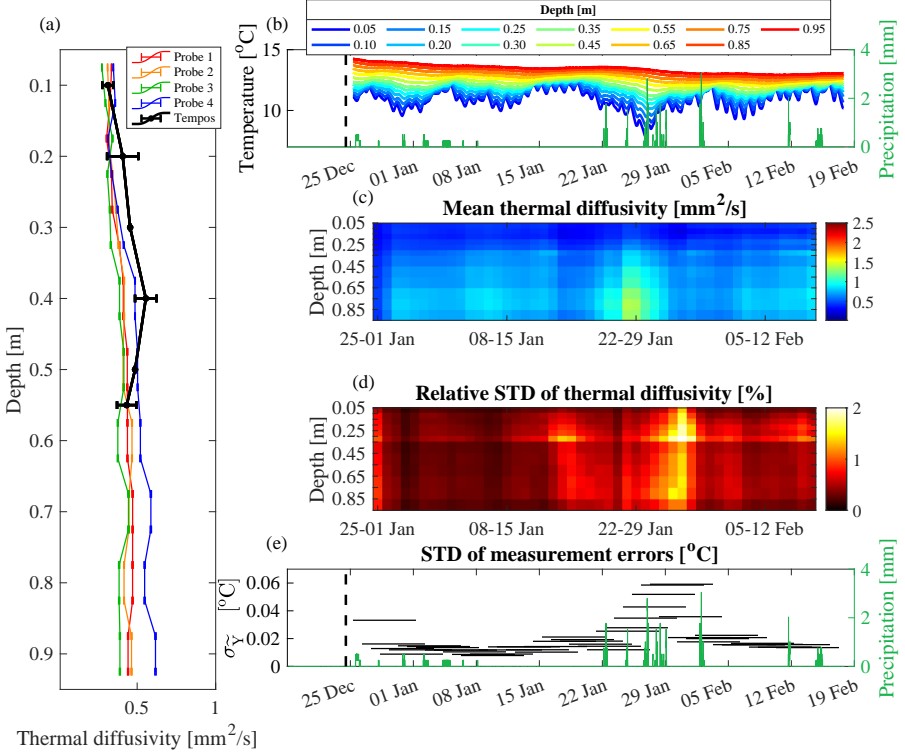

**Figure 5.** (a) Comparison of mean and standard deviation of thermal diffusivity measurements obtained with the TEMPOS thermal analyzer on 24 December 2020 (black bold line), and posterior mean and standard deviation of thermal diffusivity inferred from MCMC inversion on the temperature fields recorded from four probes (colored lines) from 25 December 2020 to 01 January 2021. (b) Temperature field recorded by Probe 1 from 25 December 2020 to 17 February 2021 and rain precipitation (green vertical bars). (c) Mean field and (d) relative standard deviation (Eq. (8)) of the MCMC derived posterior thermal diffusivity distribution for the 7 days sliding time-windows, and (e) corresponding posterior mean of the standard deviation of the measurement errors, $\sigma_{\widetilde{Y}}$.

numerous locations and compare it with independent measurements of thermal and physical properties from soil samples, in order to evaluate the strength of these relationships for potential future estimation of soil physical properties from time series of temperature.

MCMC-based estimation of thermal diffusivity every 5 to 10 cm depth is applied at 27 locations. The one-month datasets in October show very limited median diurnal fluctuations (i.e., less than 0.3 °C at 0.05 m below the ground surface) and temperature gradients are low, with values as small as 0.63 °C m$^{-1}$ at 1 m deep (more details in Sect. 2.4). Based on the results from the synthetic experiments (Sect. 3), we selected a 10-day sliding time-window and inferred thermal diffusivity up to 0.85 m depth. The level of fit achieved by the MCMC inversion reflects the accuracy of the temperature probes, since more than 370 94% of inferred data errors among all probes and all time-windows is ≤ 0.03 °C.





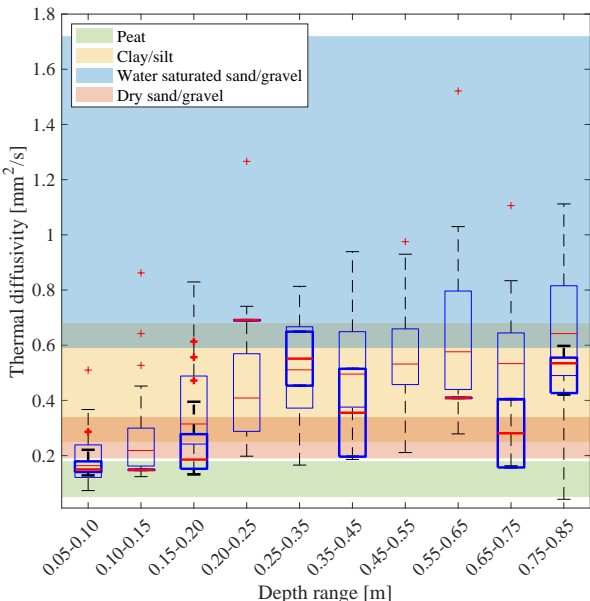

**Figure 6.** Box plots at soil depth ranges of the measured thermal diffusivity (bold boxes) from 92 soil samples and of the thermal diffusivity values (thin boxes) inferred from MCMC inversions at 27 locations. The colored shaded areas indicate the range of thermal diffusivity values that characterized different soil types (Andújar Márquez et al., 2016).

The distribution of estimated thermal diffusivity at each depth from the 27 temperature fields recorded at the Teller site (thin boxplots in Fig. 6) is consistent with the distribution of the thermal diffusivity measured from the 92 soil samples collected across the site and analyzed using the thermal analyzer (bold boxplots in Figure 6). Estimated and measured thermal diffusivities suggest overall higher values and higher variability at depth larger than about 0.2 m. The thermal diffusivity values in the

top 0.2 m are consistent with values observed for peat layer in the literature (Farouki, 1981; Andújar Márquez et al., 2016). The transition from peat to a more mineral soil occurring at a depth between 0.10 m and 0.20 m is also in agreement with the soil visual observations performed during the field campaign. Below 0.15–0.20 m, the soil is mainly composed of clay and silt (yellow shaded area in Figure 6). The correlation coefficient between measured and inferred thermal diffusivity at 20 co-located locations is 0.79. This correlation is strong, considering that (i) we are comparing thermal diffusivities measured in August

2019 with those inferred from temperature fields recorded in October 2019, (ii) thermal diffusivity from soil samples can be influenced by changes in bulk density occurring during the sampling, and (iii) the scale of the two measurements are different, with soil samples being very local, sparse, and imperfectly co-located with the temperature measurements. Moreover, we note that the temporal variability (i.e., MCMC inference from each sliding time-window over the time period of October 2019) in posterior mean thermal diffusivities from the deepest layers (not shown) is characterized by larger variability than those in the

shallowest layers. This is in agreement with the results from the synthetic experiments for which smaller temperature gradients are found at deeper depths, causing larger uncertainties in the MCMC-inferred thermal diffusivities.





We further assessed the MCMC-inferred vertically resolved profiles of soil thermal diffusivity by evaluating the relationship between measured and MCMC-inferred thermal diffusivity with other soil thermal properties (i.e., thermal conductivity and volumetric heat capacity) and physical properties (i.e., bulk density, water content, and carbon density) retrieved from

laboratory analyses (Fig. 7). For each pair of soil properties, we compute the distance correlation ($d_c$) coefficient in order to capture linear and nonlinear dependencies. The strongest distance correlation coefficient ($d_c > 0.9$) is found between thermal diffusivity (measured from soil samples and inferred with the MCMC method) and thermal conductivity, wet bulk density, and dry bulk density measured from soil samples (Fig. 7a, 7c, 7d). Also, thermal diffusivity inferred with the MCMC method and measured from soil samples are both similarly correlated ($d_c$ equals to 0.78 and 0.76, respectively) to carbon density (Fig.

7f). The lowest correlation is observed between volumetric heat capacity and measured ($d_c = 0.72$) and estimated ($d_c = 0.49$) thermal diffusivity.

A large amount of the variability in each of the thermal and physical properties is linked to the vertical heterogeneity of the soil. All the shallow soil samples within the top 0.2 m are composed of at least 50% water (Fig. 7e), are rich in organic carbon, and are characterized by lower thermal diffusivity values (Fig. 7f). Samples from depths between approximately 0.2 m and 0.5

m show a large range of variability in all the properties. Samples from deeper than 0.5 m tend to show the highest values in wet and dry bulk density and in thermal diffusivity.

## 6   Discussion

In this study, we developed a methodology for estimating soil thermal diffusivity and its uncertainty at an unprecedented vertical spatial resolution, at multiple locations, and over time, using depth-resolved time series of soil temperature, thermal modeling,

and Bayesian inference. Through synthetic and field experiments, we have assessed how various sources of uncertainty impact thermal diffusivity estimates, and we have evaluated the method's performance and its potential.

Results show that environmental conditions (i.e., temperature gradient, fluctuations, and trend) and measurement strategies affect the final amount of information contained in the temperature time series and, therefore, the quality of thermal diffusivity estimates. The outcome from our study is in agreement with the work of Rodler et al. (2019), which concluded that the higher

the temperature gradient, the better the results. In particular, we found that temperature gradients smaller than $\pm\ 2\ °C\ m^{-1}$ (Fig. 1) can be particularly problematic and lead to unreliable thermal property estimates (i.e., percentage errors larger than 30%), if a insufficiently long time-window is used. Besides the temperature gradient, another environmental condition that impacts soil thermal diffusivity estimates is the period (e.g., from daily to yearly) and amplitude of temperature fluctuations. Over a 1-month period with median diurnal fluctuations $\geq 1.5\ °C$ at 5 cm below the ground surface, temperature gradients >

$2\ °C\ m^{-1}$ at each depth and a sliding time-window of at least 4 days, the proposed method provides reliable depth-resolved thermal diffusivity estimates, with percentage error $\leq 10\%$ and posterior relative standard deviations $\leq 5\%$ up to 1 m depth. These results were obtained from challenging synthetic experiments in which the boundary conditions used for the thermal modeling are soil temperature time series recorded in Alaska where temperature gradients and diurnal temperature variations



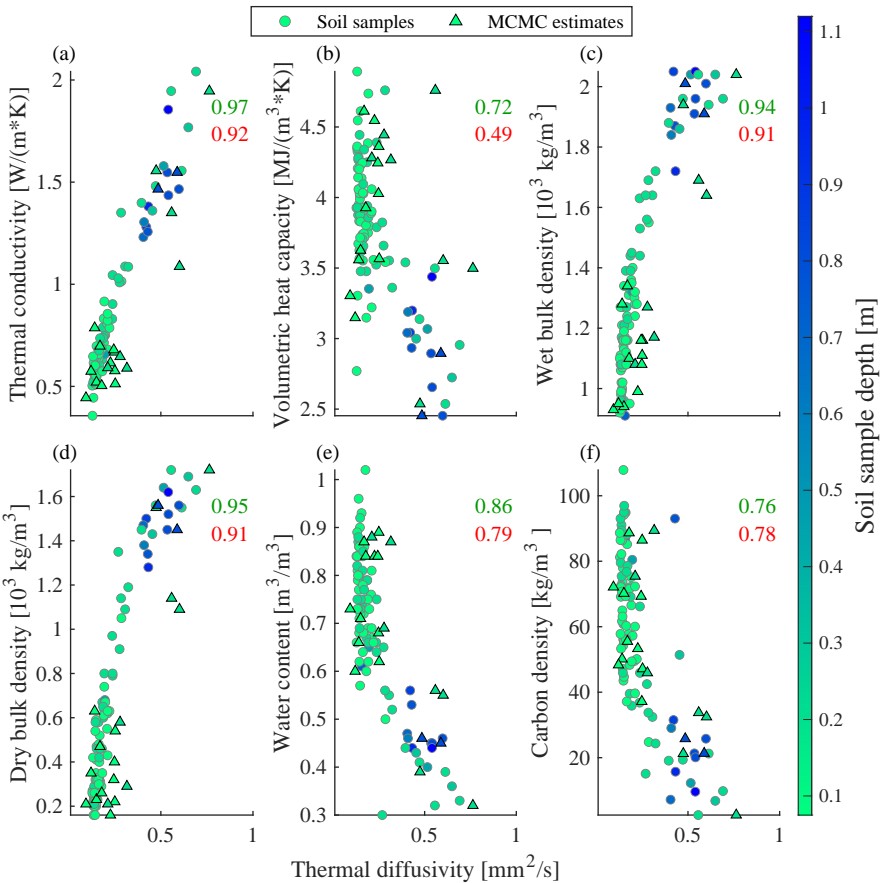

**Figure 7.** Relationship between soil thermal diffusivity and (a) thermal conductivity, (b) volumetric heat capacity, (c) wet and (d) dry bulk density, (e) water content and (f) carbon density for 92 soil samples collected across the Teller site (dots), and additional comparison with collocated MCMC-inferred (triangles) thermal diffusivity. The distance correlation between measured (green number) or MCMC-inferred (red number) thermal diffusivity and the other soil thermal-physical properties are also shown.

are typically lower than in other regions. Therefore, the proposed method has the capability to provide reliable depth-resolved
thermal diffusivity estimates at depths deeper than 1 m and over time-windows shorter than 4 days.

Low information content in the temperature time series can be partly compensated for by increasing the size of the time-window used to estimate soil thermal diffusivity. For example, by lengthening the time-window (i.e., from 7 to 20 days in our synthetic experiment), the amount of data used to compute the likelihood in the MCMC inversion increases, and we can recover reliable thermal diffusivity values up to 1 m deep under temperature gradients smaller than 2 °C m$^{-1}$ (Fig. 1h) or when diurnal
fluctuations are absent (Fig. 2d). However, we need to keep in mind that in real case studies, the time-window should be set as short as possible, to limit the influence of physical processes not represented in the heat diffusion model (e.g., advection, phase change). While incorporating flow into the model has been already achieved in the literature (e.g., Zhang et al., 2016;



Zhao et al., 2016; Tabbagh et al., 2017), it represents an additional range of complexity and might introduce additional sources of uncertainty that still need to be evaluated.

Besides environmental conditions, additional sources of uncertainty for thermal diffusivity estimates are those related to acquisition strategies, including sensor quality, calibration, and deployment geometry. In Sect. 3.3, we investigated the impact of temperature fields perturbed with different levels of noise and bias. Results have shown that the use of sensors with even a small amount of bias is more detrimental to thermal diffusivity estimates than having sensors with some random noise (level of noise $\geq 0.05$ °C), because the former might ultimately lead to a misleading interpretation of the soil layering (Fig. 3a–d vs

Fig. 3g–j). Indeed, the availability of high-accuracy sensors, with a bias defined by a standard deviation of 0.01 °C or less, is fundamental to ensure percentage errors on thermal diffusivity $\leq 10\%$. This accuracy requirement underlines the challenge of this method and the importance of sensor-calibration approaches to increase sensor accuracy as much as possible.

Furthermore, results show that the deployment geometry is critical to potentially capturing heterogeneity present in the soil, and particularly the layer boundaries. The heterogeneity underlines the importance of inferring soil diffusivity with high spatial

resolution, as done in this study. Note that the impact of sensor geometry on thermal diffusivity estimates is expected to be even larger in more heterogenous soils. Finally, while several environmental factors and measurement strategies strongly influence estimate accuracy, the potential error in mispositioning a sensor in soil, which is less likely using a temperature probe than individual sensors, can easily lead to percentage errors in thermal diffusivity estimates larger than 30% (Fig. 4). We conclude that, under the environmental conditions described above (i.e., median diurnal fluctuations $\geq 1.5$ °C, temperature gradients $>$

$2$ °C m$^{-1}$), temperature sensors with a level of noise $\leq 0.02$ °C, a bias defined by a standard deviation of 0.01 °C or less, and a positioning accuracy of a few millimeters or less (i.e., temperature probe) is needed to ensure reliable thermal diffusivity estimates up to 1 m deep.

The reliability of the developed approach has been demonstrated for in situ estimation of vertically resolved thermal diffusivity profiles at a site in Berkeley (CA, USA) and at Teller site, northwest of Nome (AK, USA). At both sites, the estimated

values compare well with independent measurements obtained using a thermal properties analyzer. Still, they do not match perfectly, because they are measured at different spatial scales. Moreover, soil temperatures recorded at Teller site in autumn are characterized by very low temperature gradients and diurnal fluctuations which make thermal diffusivity estimates extremely challenging. Overall, the obtained thermal diffusivity values were consistent with those reported in the literature for the various soil types encountered at these sites.

The field results underline the value of thermal diffusivity in quantifying soil physical properties, as well as challenges associated with interpreting spatial and temporal variations in thermal diffusivity. For example, the increase in water content in a particular dry material ($< 15\%$) can drive an increase in thermal diffusivity, while a similar increase in a wetter material ($> 15\%$) will barely impact the thermal diffusivity or drive a decrease in it (Farouki, 1981). This complexity explains why we observe a negative correlation between thermal diffusivity and soil moisture at the site along Teller Road, AK (Fig. 7e), while

the thermal diffusivity of deep soil at the Berkeley site increases after a rain event (Fig. 5). In the latter case, the soil was drier in the deep subsurface as a consequence of a long dry period that lasted about 1 month.





The importance of estimating thermal diffusivity lies in the fact that this soil property shows strong correlation (i.e., between 0.78 and 0.92) with thermal conductivity, wet and dry bulk density, water content, and carbon density (Fig. 7). These results confirm previous laboratory studies (Farouki, 1981; Arkhangelskaya and Lukyashchenko, 2018; Mengistu et al., 2017; Ochsner and Baker, 2008) investigating these relationships and offer new datasets to improve the integrated estimation of these properties. The recent work from Zhu et al. (2019) highlights the value of temperature measurements in quantifying thermal diffusivity and further evaluating the dominant effect of organic carbon on permafrost dynamics.

Finally, results from this study show promise in using temperature time series to estimate temporal changes in soil diffusivity. This is the first time to our knowledge that this concept has been introduced in environmental science at such a vertical spatial resolution and in a Bayesian framework. Tabbagh et al. (2017) has used a similar sliding time-window approach to evaluate the temporal variability of thermal diffusivity, but for a single layer and with a deterministic inverse method. The assumptions of our method, based on which thermal diffusivity is constant in each time-window, and on purely heat-conduction processes taking place, can be violated if soil wetness changes considerably and if time-scales are smaller than the length of the time-window. However, we can still detect these time periods by looking at the MCMC outputs and at the inferred data errors, as shown in this study: the precipitation event at the Berkeley site caused at least a twofold increase in the MCMC-inferred thermal diffusivity estimates (Fig. 5c) and corresponding posterior relative standard deviations (Fig. 5d), as well as a larger inferred data error (Fig. 5e).

## 7 Conclusions

In this work, we have presented a parameter estimation approach based on the combination of thermal modeling (i.e., heat diffusion equation), sliding time-windows, Bayesian inference, and MCMC simulation. The method enables us to estimate soil thermal diffusivity and its uncertainty from solely depth-resolved temperature time series at an unprecedented vertical spatial resolution (i.e., 5 to 10 cm resolution up to 1 m deep), at multiple locations, and over time. By taking into account various sources of uncertainty in our method, we were able to provide the necessary framework and synthetic test cases to assess under which environmental conditions (soil temperature gradient, fluctuations, and trend) temperature sensor characteristics (bias and level of noise), and deployment geometries (number and relative position) soil thermal diffusivity can be reliably inferred. In addition, the application of the developed approach to field data indicates significant repeatability in results and consistency of the estimated soil thermal diffusivities with independent measurements as well as with values reported in the literature for the various soil types encountered at the two sites considered in this work. The field studies show also promise in using a sliding time-window to estimate temporal changes in soil thermal diffusivity which can be used to capture the corresponding changes in carbon or water content.

The value of our methodology lies in the fact that thermal diffusivity can be quantified from depth-resolved temperature time series, at numerous locations, using Distributed Temperature Profiling systems. This is of great value, considering that thermal diffusivity is strongly linked with thermal conductivity, wet and dry bulk density, water content, and carbon density, as shown by this and previous works. Hence, our approach opens the way for future research on the development of petrophysical



relationships that can be integrated within the Bayesian inversion framework, and used to derive from soil thermal diffusivity, inferred from temperature time series, the fraction of soil components that is fundamental to a better understanding of the subsurface storage and fluxes of water, carbon, and nutrients under a warming climate. Temperature time series data have also been proven to contain valuable information on snow thermal properties, and the proposed methodology could also be applicable to this purpose.

*Data availability.* Soil temperature time series used in the synthetic experiments in this study were deposited in the Zenodo repository (https://doi.org/10.5281/zenodo.5465253).

*Author contributions.* Carlotta Brunetti: Formal analysis, methodology, validation, visualization, writing, data collection. John Lamb: Data collection. Stijn Wielandt: Sensor development. Sebastian Uhlemann: Data collection. Ian Shirley: Data collection. Patrick McClure: Sensor development. Baptiste Dafflon: Conceptualization, Funding acquisition, review and editing, data collection

*Competing interests.* The authors declare that they have no conflict of interest.

*Acknowledgements.* This research has been supported by the Office of Biological and Environmental Research in the DOE Office of Science (grant no. DE-AC02-05CH11231). We thank Vladimir Romanovsky for providing soil temperatures used for the synthetic examples. We thank Jasper Vrugt for making DREAM$_{(ZS)}$ available by providing an initial version of the code.





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
