# Peer review of "Probabilistic estimation of depth-resolved profiles of soil thermal diffusivity from temperature time series"

_Earth Surface Dynamics, 2021_

## Author Response (AR1)

In this document, you can find the questions from the reviewers, the corresponding responses of the authors in *italics* and at which line number the authors have changed the manuscript. Note that the number of the lines refer to those in the manuscript with tracked changes.

**REVIEWER 1**

The manuscript describes an interesting and timely study that applies the Bayesian method to the measured at multiple depth temperature time-series to estimate/recover thermal diffusivity. The method and implementation are sound. My main concerns:

1. The choice of the model. The thermal model worked well for the examples shown in this study (i.e. when we are dealing with temperatures above 0∘C). However, it was shown in many studies that applying the classical thermal equation described in this study does not capture the full temperature dynamics in permafrost-affected soil (Romanovksy et al., 2000). The effect of the unfrozen water is an essential factor that needs to be accounted for.

*Dear reviewer,*
*Thank you very much for taking the time to review the manuscript and provide your valuable feedback.*
*Based on the reviewers' comments, we realized that we did not explain clearly enough the reasons why in this study we are using a heat-conduction-based model without representation of advection and latent heat process, its advantages and limitations, and why we used it in an Arctic environment.*

*The thermal model we are using (heat-conduction-based model without representation of advection and latent heat process) is only adequate and used to simulate heat transfer during periods of time when conduction dominates advection and when temperature remains well above 0∘C (above 0.5∘C in our study). This means that in our case study in the Arctic environment, we only apply this model to time periods when the soil is entirely unfrozen along the probe and when rain events are minimal. We agree that estimating thermal diffusivity during freeze-thaw processes would require us to account for the effect of unfrozen water (*Romanovksy et al., 2000*). Still, we do not intend to do that in our study.*

*The main reason that guided our choice of using the heat-conduction-based model is that in many situations, hydrological boundary conditions are difficult to assess. Indeed, using a more complex model (including advection and/or latent heat) would require to force or parametrize the model with information that is much more difficult to collect than temperature. For example, rain precipitation and its partitioning between surface flow and infiltration is a major source of uncertainty that we would need to account for if we simulate soil moisture variations. Similarly, representing freeze-thaw process would require to assess first the water content in unfrozen soil and thus would require to measure or estimate several parameters. While we are aware that a few studies have been estimating thermal conductivity by including the freeze-thaw process, they have generally assumed the total water content constant, used additional datasets from intensive sites, or made assumptions given the site hydrological conditions (e.g., full saturation) (Jafarov et al., 2014, Nicolsky et al., 2010). Still, in our study we aimed at developing a method that can be applied at locations where only temperature data are present, and where soil water saturation and porosity is variable. Thus, we decided to use a heat-conduction-based model without representation of advection and latent heat process to reduce the number of parameters to be estimated or constrained (and hence the likelihood of non-uniqueness of solutions) while limiting the*

*applicability of our model to time periods when the soil is unfrozen and the heat advection is limited (i.e., dry period considered for our case study in the Arctic environment).*

*Finally, it can be noted that our approach involves repeating the estimation of thermal diffusivity with a moving time-window that:*
*1- is as short as possible in order to enable the estimation of thermal diffusivity during time-windows devoid of significant water fluxes (e.g., advection) not represented in our heat-conduction-based model, as well as the detection of advection processes that may occur intermittently e.g., caused by percolation events. In this study (Section 4), we showed that when such events take place the misfit between the model and the data is expected to increase, and thus the moving time-window approach can be used to evaluate when our model lack hydrological process representation.*
*2- does enable the detection of changes in thermal diffusivity over time, presumably linked to changes in water content*
*Overall, while we agree that our model does not represent all the complexity of hydrological processes, we believe this approach is pragmatic for many applications, and reliable when applied under the above-mentioned conditions.*

*We clarified this in the revised manuscript (lines 95–102 and 531-538 in the manuscript with tracked changes).*

2. How the Bayesian approach used in this study is different from the variational approach when the gradient is explicitly/implicitly calculated to find the next iteration?

*The bayesian inference problem can indeed be solved using various methods among which the Markov chain Monte Carlo (MCMC) method (as applied in our study) or the variational approach.*

*The main difference is that the MCMC method approximate the target (posterior) distribution via sampling schemes (i.e., no model is assumed) whereas variational inference finds the best parametrized ensemble of distributions that represent the target one (i.e., a model is assumed). Generally, MCMC methods are more accurate than variational approaches but require more computational resources. (More info can be found here: Blei, David M., Alp Kucukelbir, and Jon D. McAuliffe. "Variational inference: A review for statisticians." Journal of the American statistical Association 112.518 (2017): 859-877).*

3. Please extend the discussion on estimating thermal diffusivities at or near 0ₒC and estimating thermal diffusivities during time periods when the gradient sign changes from positive to negative.

*In our study, we use only temperature time series that are at all depths above 0.5ₒC (soil entirely unfrozen). Indeed, close to 0ₒC latent heat transfer occurs due to water freezing and melting processes. In this study, we infer thermal diffusivity only in time-windows in which temperature is above 0.5ₒC in order to not violate our conductive heat transfer model assumption.*

*By applying our inference approach under various temperature fields, we observed that the change in sign of the temperature gradient (i.e., similar to the case in Fig 1 in the paper) does not cause any issue in the reliable inference of thermal diffusivity by itself. However, if the temperature*

*gradient changes sign very slowly over time then it likely implies low temperature gradients over long period of time which will degrade the inference of the thermal diffusivity.*

*We clarified this in the revised manuscript (lines 95–102 in the manuscript with tracked changes).*

4. In Section 3.4. it was not clear how the optimal number of sensors had to be selected. In the Discussion, the authors mentioned the effect of low gradients between temperature time-series makes it hard to estimate diffusivity. This finding is also consistent with studies by Jafarov et al., (2014; 2020). What is the best way to proceed in the case of a low gradient? Is it better to exclude those chunks of data from the method? How can this method be used at the design stage to build temperature probes that could capture optimal temperatures signals (i.e. no low gradient signals)?

*Thank you for mentioning Jafarov et al., (2014; 2020) that we have now cited in the discussion section.*
*In our view, there are two possible way to proceed in the case of low temperature gradients:*
- *remove the time windows with low temperature gradients from the analysis*
- *increase the length of the moving time window and/or decrease the depth at which infer thermal diffusivity (e.g., for the application of our method in the Teller site in Alaska, we decided to increase the time window from 7 to 10 days and to investigate thermal diffusivity up to 0.85 m instead of 1m). We improve the discussion on the time windows length in the revised manuscript.*

*We clarified this in the revised manuscript (lines 401-403, 467-468 in the manuscript with tracked changes).*

*Thank you for your interesting question on the creation of the temperature probe to optimally capture temperature signals.*
*Given the high variability of the temperature gradients overt time and of soil composition, it might be difficult to think of a probe at the design stage that is generally optimized for low gradient signals. Based on the synthetic experiments implemented in section 3.4, we were however able to identify some important aspects to consider at the design stage of temperature sensors such as:*
- *having more sensor closer to the soil surface where temperature signal has the highest content of information (e.g., diurnal and seasonal fluctuations).*
- *using temperature probes (i.e., sensors anchored on the same support as used in this study) instead of discrete temperature sensors placed manually in the subsurface, which implies higher uncertainty in the distance between sensors.*
- *ensuring a vertical spatial resolution that captures as much as possible the soil layering and thus decreasing the sensor spacing in vertically heterogeneous environments*
- *collecting high-accuracy measurements, which require optimization in sensor accuracy, sensor spacing and the all system accuracy. Improving system accuracy is still challenging, yet critical depending on the environmental conditions.*
- *taking into consideration the environmental conditions at the sites of interest*

*In addition, we provided values that one can consider as a starting point to design a probe optimal for their study. Indeed, we concluded that, under the environmental conditions with median diurnal fluctuations $>= 1.5^oC$, temperature gradients $> 2^oC\ m^{-1}$, temperature sensors with a level of noise $<0.02^oC$, a bias defined by a standard deviation of $0.01^oC$ or less, and a positioning accuracy of a few millimeters or less (i.e., temperature probe) is needed to ensure reliable thermal diffusivity estimates up to 1 m deep.*

*We added these points in the revised manuscript (lines 484–495 in the manuscript with tracked changes).*

Minor concerns:

Consider simplifying the title and removing "and uncertainty quantification" because, in the end, you recovered diffusivity from temperatures not from UQ. Consider including UQ into the search words for the paper.

*We modified the title to "Probabilistic estimation of depth-resolved profiles of soil thermal diffusivity from temperature time series", see pag. 1 in the manuscript with tracked changes.*

L121-122 It was mentioned that diffusivity depends on soil moisture. It is not clear how water content is accounted for in the eqn. 2.

*The fraction of water in soil affects the density, the specific heat capacity and the thermal conductivity. Thermal diffusivity depends on all these three quantities and therefore it is indirectly affected by the soil moisture.*

*We clarified this in the revised manuscript (lines 138–139 in the manuscript with tracked changes).*

L421-422 The moving-time window. Jafarov et al., (2014) found that 30 days moving average filter worked the best when applied to changing over time snow thermal conductivity. Will 30 days average window here? If not, why?

*The optimal length of the time-window depends on the data information content and the site-dependent nature of hydrological processes. Comparing our time-window strategy to the 30 days moving average filter in Jafarov et al., (2014) does not seem adequate to us. Jafarov et al. (2014) study, though very interesting in many aspects, seem to use a time-window to constrain the variability of snow thickness. We think this goal is quite different from our objective when using a moving time-window.*

*In our study, we used a time-window length as short as possible in order to limit the influence of hydrological processes (e.g., water fluxes not represented in the heat-conduction-based model) but sufficiently long to reliably infer thermal diffusivity. Moreover, the shorter the time-window the higher the temporal resolution at which thermal diffusivity changes can be detected. In our study, we use 7 days (10 days for the study at Teller site) moving time-window. However, we show that 4 days are sufficient under certain environmental conditions (Section 3.1, Figure 1).*

In conclusion, include the caveats about the model (does not account for soil moisture), the method (will it fail with low gradient data), and how choosing different moving average windows will affect the estimated diffusivity.

*We added sentences in the introduction, result and discussion sections to state more clearly the reasons why we use a heat-conduction-based model and the related limitations of our approach. Concerning the impact of choosing different lengths for the time window, we clarified it in Section 3.1, depicted in Figure 1 and discussed in Sec. 6.*

*We have added all these changes in the revised manuscript (lines 95–102, lines 292–294 , 401-403, 464-468, 495-499, and 531-538 in the manuscript with tracked changes).*

**REVIEWER 2**

The paper « Estimation of depth-resolved profiles of soil thermal diffusivity from temperature time series and uncertainty quantification » presents a modeling study of near-surface thermal diffusivity of soil using the Bayesian method and validates its approach with synthetic experiments and field data collected at a site in Alsaka.

**General comments**

The study is of high interest and the paper is very well written. Several points could be improved to help the paper gaining a broader impact and to make it clearer to a diverse audience such as one could expect with *Earth Surface Dynamics*. While the introduction shows efforts in clarifying the various modeling approaches, the authors are sometimes too straightforward on some aspects and may disconcert a part of the possible readers.

Field data and field site would deserve a more thorough description. What kind of ground it is? What is the period of measurement? Which sensors are used? What are the weather records during this period? etc. The connexion between modeling experiments and field data is quite unclear. The author thank Vladimir Romanovsky for providing data but do not explain why they chose this dataset. Details on the soil characteristics, lithology, climate characteristics, the data collection approach, the choice of the dataset, the data characteristics, etc. must be provided.

*Dear reviewer,*
*Thank you very much for taking the time to review the manuscript and provide your valuable feedback.*

*We modified the manuscript to provide more details on the measurements we used to create synthetic dataset. The measurements were acquired at a site on the Seward Peninsula, Alaska during the summer (Romanovsky et al., 2020) and the autumn ([https://doi.org/10.5281/ zenodo.5465253](https://doi.org/10.5281/zenodo.5465253)) period. These datasets are from locations characterized by the absence of permafrost, the presence of tall shrubs, and decreasing organic matter content and porosity with depth. Thermal diffusivity values of the three layer model were assigned based on soil sample analysis performed at the same site under similar environmental conditions.*

*We clarified this in the revised manuscript (lines 222-216 in the manuscript with tracked changes).*

*We also clarified the link between synthetic experiments and field case studies in the introduction. Indeed, we perform synthetic experiments in which we infer soil thermal diffusivity and assess its uncertainty under different environmental conditions (i.e., soil temperature gradients and fluctuations), length of sliding time-window, level of measurement errors, and temperature sensor geometries (Sect. 3). In Sect. 4 we further evaluate the reliability and sensitivity of the proposed method with an in situ study that compares estimated thermal diffusivities for a silty/clayey soil in a warm summer mediterranean climate (Berkeley, California, USA) with independent measurements obtained with a thermal-properties analyzer. Finally, in Sect. 5, we apply the method at a field site in a discontinuous permafrost environment in a tundra climate (Nome, Alaska, USA) during a*

*period of low vertical gradient in soil temperature, and compare thermal diffusivity estimates at numerous locations across the site with soil sample measurements. The synthetic and in situ experiments enable to evaluate and validate the developed method and its applicability to numerous environmental conditions, including challenging situations such as the presence of low vertical gradients in soil temperature. We clarified this in the revised manuscript (lines 111-120 in the manuscript with tracked changes).*

*We also clarified the link between synthetic experiments and field case studies by changing the titles of section 2.4.3, 2.4.4, 4 and 5.*

The broad significance of the study is also overlooked. Same words are repeated throughout the manuscript to point out the broad significance (e.g. carbon and water fluxes) but more details or discussion points against examples would have more impact.

*We improved several sentences in the manuscript to be more precise on how the presented approach can help improve the carbon and water fluxes. Specifically, the evaluation of our method in an Arctic environment is particularly important to potentially improve the parameterization of soil thermal parameters and organic matter content in ecosystem models simulating the feedback from Arctic ecosystem to climate warming. We clarified this in the revised manuscript (lines 121-123 in the manuscript with tracked changes).*

*And also, our approach opens the way for future research on the development of petrophysical relationships that can be integrated within the Bayesian inversion framework, and used to derive from soil thermal diffusivity, inferred from temperature time series, the fraction of soil components, and in particular carbon density in organic rich environments. The carbon density is a key property in the quantification of soil respiration rate. Similarly, an improved quantification of soil thermal parameters is needed to better parametrized ecosystem models, where soil thermal parameters modulate the effect of weather forcing on subsurface thermal and biogeochemical fluxes. We clarified this in the revised manuscript (lines 565-571 in the manuscript with tracked changes).*

The interest of this study is the reproducibility of the calculation and the tools and codes for these calculations are not provided. Please, provide more details or data for reproducing the study on other ground and if relevant provide the code. Otherwise explain why it is not provided.

*We have deposited the forward model written in Matlab that compute the 1D diffusion equation in an heterogeneous medium according to Equation 3 in the paper at https://doi.org/10.5281/zenodo.6350359. The script provided has been validated (i.e., tested with synthetic data generated by analytical calculation). The code on the inverse model based on MCMC and written in Matlab is a proprietary software (https://www.pc-progress.com/en/Default.aspx?dream ). However, similar implementation has been written in python and freely accessible here https://github.com/LoLab-VU/PyDREAM . All this information has been reported in the "Code availability statement" at the end of the paper (lines 572-575 in the manuscript with tracked changes).*

One crucial point remains unclear to me: the field data are collected in a permafrost ground but the modeling approach is not appropriate for permafrost modeling: the latent heat is ignored (see comment below) and freezing processes are not discussed. Furthermore results show positive temperature. These limitations raise a major concern about the modeling approach that encourages to reconsider the paper after major revisions if the authors can not clarify their approach.

*Based on the reviewers' comments, we realized that we did not explain clearly enough the reasons why in this study we are using a heat-conduction-based model without representation of advection and latent heat process, its advantages and limitations, and why we used it in an Arctic environment. We clarified this in the revised manuscript.*

*The thermal model we are using (heat-conduction-based model without representation of advection and latent heat process) is only adequate and used to simulate heat transfer during periods of time when conduction dominates advection and when temperature remains well above 0ₒC (above 0.5ₒC in our study). This means that in our case study in the Arctic environment, we only apply this model to time periods when the soil is entirely unfrozen along the probe and when rain events are minimal. We agree that estimating thermal diffusivity during freeze-thaw processes would require us to account for the effect of unfrozen water (Romanovksy et al., 2000). Still, we do not intend to do that in our study.*

*The main reason that guided our choice of using the heat-conduction-based model is that in many situations, hydrological boundary conditions are difficult to assess. Indeed, using a more complex model (including advection and/or latent heat) would require to force or parametrize the model with information that is much more difficult to collect than temperature. For example, rain precipitation and its partitioning between surface flow and infiltration is a major source of uncertainty that we would need to account for if we simulate soil moisture variations. Similarly, representing freeze-thaw process would require to assess first the water content in unfrozen soil and thus would require to measure or estimate several parameters. While we are aware that a few studies have been estimating thermal conductivity by including the freeze-thaw process, they have generally assumed the total water content constant, used additional datasets from intensive sites, or made assumptions given the site hydrological conditions (e.g., full saturation) (Jafarov et al., 2014, Nicolsky et al., 2010). Still, in our study we aimed at developing a method that can be applied at locations where only temperature data are present, and where soil water saturation and porosity is variable. Thus, we decided to use a heat-conduction-based model without representation of advection and latent heat process to reduce the number of parameters to be estimated or constrained (and hence the likelihood of non-uniqueness of solutions) while limiting the applicability of our model to time periods when the soil is unfrozen and the heat advection is limited (i.e., dry period considered for our case study in the Arctic environment).*

*Finally, it can be noted that our approach involves repeating the estimation of thermal diffusivity with a moving time-window that:*
*1- is as short as possible in order to enable the estimation of thermal diffusivity during time-windows devoid of significant water fluxes (e.g., advection) not represented in our heat-conduction-based model, as well as the detection of advection processes that may occur intermittently e.g., caused by percolation events. In this study (Section 4), we showed that when such events take place the misfit between the model and the data is expected to increase, and thus the moving time-window approach can be used to evaluate when our model lack hydrological process representation.*
*2- does enable the detection of changes in thermal diffusivity over time, presumably linked to changes in water content*
*Overall, while we agree that our model does not represent all the complexity of hydrological processes, we believe this approach is pragmatic for many applications, and reliable when applied under the above-mentioned conditions.*

*We clarified this in the revised manuscript (lines 95–102 and 531-538 in the manuscript with tracked changes).*

**Introduction :** The introduction is very well written and provides an interesting overview of existing modeling approach.

L 89-90 : What does « numerous » means? What does « landscape » means when it is written later that all data are collected very close to each other in the same type of ground? The data are obviously not collected at a « landscape scale » but at a sub-hectometric or similar scale in homogeneous terrain. This remark points out the need to introduce the field site in more details.

*"Numerous" locations refer to the locations at which the temperature probes were inserted in the soil to record temperature time-series (up to 27 locations over 2.3 km² at Teller site in Alaska). We have added a sentence on this in the paper. We have also removed the word "landscape" and replaced with "field site" where appropriate in order to avoid confusion (see lines 32, 90, 109, 261-262 and 502 in the manuscript with tracked changes)*

Similarly, about questions (1) and (3), I do not understand how the study tackles the question of « different environmental conditions » or « different locations across the landscape » by focusing on a single site, even though several dataset are collected at this site.

*We have clarified these points in the introduction. The field applications to different environmental conditions is aimed to testing the method under different soil temperature gradients and fluctuations which reflect different weather forcing. Indeed, we apply our method to two different sites: one in a warm mediterranean climate in California (Sec. 4) and one in a tundra climate in Alaska (Sec. 5).*

*The field site study in California is used to:*
*1- test the repeatability of our approach (the soil temperature is measured multiple times within a 25 cm radius area)*
*2- test the ability of our method to detect changes of thermal diffusivity over time*

*The field site study in Alaska is used to:*
*1- test the ability of our method to detect changes of thermal diffusivity over time and space (soil temperature time-series collected from 27 location within the 2.3 km² area of the Teller site)*
*2-evaluate the link between estimated soil thermal properties and soil physical properties from soil samples.*

*We clarified this in lines 111-123 in the manuscript with tracked changes and by changing the titles of section 2.4.3, 2.4.4, 4 and 5.*

**Theory and method**

The approach contains many steps. I would suggest that the authors add a diagram outlining the different steps and how they interact to make it clearer (section 2 and 3).

*We added a clear description of the various experiments and their goals (see response to previous comment) in the introduction. We prefer to not add a figure given that most of this information can be provided with words and that there is no iterative feedback mechanisms between these various experiments.*

L 110-115: are the latent heat exchanges ignored in the modeling approach? It is a crucial process in permafrost ground and it has to be accounted for!

*We did not apply our method to soil temperature <0.5 ºC (see our response to earlier comment on field application in a discontinuous permafrost system). We have already addressed this question above.*

Section 2.4.2: how is the thermal diffusivity assumed? Based on which knowledge?

*Thermal diffusivity values are chosen based on the values measured from laboratory analysis on soil samples collected at the same site and under similar environmental conditions (see line 231 in the manuscript with tracked changes).*

**Results**

I think that the results of the soil samples analysis outlined in section 2.4.4 should be clearly described such as the modeling results. This would also help to link modeling experiments and field data.

*We added more details on the soil laboratory analysis in the manuscript (lines 268-272 in the manuscript with tracked changes). Laboratory analysis involved drying the soil samples at 65 ºC, recording dry weight, grind soil to pass a 2mm sieve, record weight of the >2mm and <2mm portion, and measure carbon concentration by combustion.*

**Discussion**

It would be easier to follow with subsections. It seems that many paragraphs repeat each other and that the ideas are not well ordered.

*We have restructured the discussion section (lines 444-499 in the manuscript with tracked changes) and added the following subsection to make it easier to follow:*
*- Impact of environmental conditions, sensor characteristics and deployment geometry on thermal diffusivity estimates (line 442 in the manuscript with tracked changes)*

*- Spatial variability of thermal diffusivity and link with soil physical properties (line 500 in the manuscript with tracked changes)*

*- Temporal variability of thermal diffusivity (line 521 in the manuscript with tracked changes)*

**Conclusion**

It must be more specific with clear statements about the findings and broad significance of the results. In the current state it rather looks like paraphrasing of general statements provided in previous sections.

*We modified and added clear statements on the results in the first paragraph (lines 544-552 in the manuscript with tracked changes) and discuss their relevance in the second paragraph (lines 552-571 in the manuscript with tracked changes).*

I am still not convinced about the use of wording such as « various locations » knowing that all data come from the same site.

*We have addressed this concern above.*

**Detailed comments**

L 96: why is this 10% or 5% threshold relevant? Maybe providing examples of possible applications with such thresholds would be relevant.

*These thresholds are relevant because is the accuracy at which thermal diffusivity can be at best be measured from the thermal analyzers on the market such as the TEMPOS instrument with the SH-3 dual needle from the METER Group. We have added a sentence in the revised manuscript to clarify this (lines 104-105 in the manuscript with tracked changes).*

L 137-138: on which basis these diffusivity values were determined?

*The Courant-Friedrichs-Levyor condition is computed considering the maximum value of thermal diffusivity that could be used in the forward modelling. In order to set this upper limit for soil thermal diffusivity, we have reviewed the literature and found that the typical largest value is about $3 \ mm^2 s^{-1}$ (e.g., Farouki (1981); Andújar Márquez et al. (2016)). We have now clarified this in the revised manuscript (lines 153-154 in the manuscript with tracked changes).*

L 142: some data are mentioned but they should be introduced earlier in the paper.

*In Section 2 ("Theory and method"), we do not provide all the details about the data used since they are not needed at this stage. Instead, detailed info on the data are provided in Section 3 on "Data and field sites description".*

L 198: what reference/manufacturer are those sensors from?

*The digital temperature sensors (TMP117AIDRVR, http://www.ti.com/lit/ds/symlink/tmp117.pdf ) are mounted on interconnected printed circuit board and inserted in a ~ 10 mm outer diameter (OD) plastic tube filled with epoxy (Dafflon et al., 2022). We added more details in the revised manuscript (lines 211-213 in the manuscript with tracked changes).*

L 207-208: on which basis are these values determined?

*Thermal diffusivity values mentioned in Section 2.4.2 and used in the synthetic experiments are chosen based on the values measured from laboratory analysis on soil samples collected at the same site and under similar environmental consitions. A sentence has been added in the revised manuscript to clarify this (lines 231-232 in the manuscript with tracked changes).*

Fig. 1: what is the « true » thermal diffusivity ? The one calculated from field samples (Sect. 2.4.4)?

*The "true" thermal diffusivities here refer to the values derived from soil samples and used to generate the synthetic temperature fields used in the synthetic experiments.*

L 333: what are these visual observations?

*The visual observations refer here to what was observed from the soil pit dug at the site. We have clarified this in the revised manuscript (line 366 in the manuscript with tracked changes).*